# Individual Variability in Brain Connectivity Patterns and Driving-Fatigue Dynamics

**DOI:** 10.3390/s24123894

**Published:** 2024-06-16

**Authors:** Olympia Giannakopoulou, Ioannis Kakkos, Georgios N. Dimitrakopoulos, Marilena Tarousi, Yu Sun, Anastasios Bezerianos, Dimitrios D. Koutsouris, George K. Matsopoulos

**Affiliations:** 1Biomedical Engineering Laboratory, National Technical University of Athens, 15772 Athens, Greece; ogiannakopoulou@biomed.ntua.gr (O.G.); dkoutsou@biomed.ntua.gr (D.D.K.); gmatsopoulos@biomed.ntua.gr (G.K.M.); 2Department of Biomedical Engineering, University of West Attica, 12243 Athens, Greece; 3Department of Informatics, Ionian University, 49100 Corfu, Greece; 4Key Laboratory for Biomedical Engineering of Ministry of Education of China, Department of Biomedical Engineering, Zhejiang University, Hangzhou 310027, China; 5Brain Dynamics Laboratory, Barrow Neurological Institute (BNI), St. Joseph’s Hospital and Medical Center, Phoenix, AZ 85013, USA; tasso.bezerianos@commonspirit.org

**Keywords:** mental fatigue, driving, EEG, Phase Lag Index (PLI), brain networks, frequency bands

## Abstract

Mental fatigue during driving poses significant risks to road safety, necessitating accurate assessment methods to mitigate potential hazards. This study explores the impact of individual variability in brain networks on driving fatigue assessment, hypothesizing that subject-specific connectivity patterns play a pivotal role in understanding fatigue dynamics. By conducting a linear regression analysis of subject-specific brain networks in different frequency bands, this research aims to elucidate the relationships between frequency-specific connectivity patterns and driving fatigue. As such, an EEG sustained driving simulation experiment was carried out, estimating individuals’ brain networks using the Phase Lag Index (PLI) to capture shared connectivity patterns. The results unveiled notable variability in connectivity patterns across frequency bands, with the alpha band exhibiting heightened sensitivity to driving fatigue. Individualized connectivity analysis underscored the complexity of fatigue assessment and the potential for personalized approaches. These findings emphasize the importance of subject-specific brain networks in comprehending fatigue dynamics, while providing sensor space minimization, advocating for the development of efficient mobile sensor applications for real-time fatigue detection in driving scenarios.

## 1. Introduction

Driving is a multifaceted and exceptionally demanding activity carried out on a daily basis by numerous individuals worldwide. Operating a vehicle necessitates drivers to proficiently perceive and comprehend various aspects related to their driving capabilities, the condition of the driver, vehicle performance, and traffic dynamics [1]. This entails maintaining attentiveness and cognitive sharpness while rapidly processing multiple sources of information to navigate safely and effectively [2]. Conversely, extended periods of driving can induce mental fatigue, which refers to a condition wherein heightened mental strain and drowsiness reduce the driver’s capacity to respond efficiently to unforeseen or critical circumstances, thereby compromising driving safety [3].

In general, mental fatigue is a cognitive state resulting from prolonged mental activity, characterized by a decline in cognitive performance and increased feelings of tiredness and lack of energy. It has been shown to significantly impact various cognitive functions, including attention, working memory, and executive function [4].

To evaluate fatigue during driving, many strategies have been proposed, including estimation through subjective measures (such as self-reported questionnaires) [5] or physiological examination of mental fatigue (heart rate variability and skin electrical potential) [6]. Other approaches include the estimation of the brain electrical signals (via electroencephalography, EEG) for the purpose of measuring and identifying drivers’ fatigue [7]. It should be noted that although self-perceived assessment usually fails to overcome subjective perception concerning the individuality of the subjects [8], utilizing physiological measures can provide insights into the actual impact of mental fatigue on cognitive functioning, independently of individuals’ subjective perceptions [9]. In fact, cortical variability in states of fatigue is well-documented across multiple studies, demonstrating significant differences in how individuals’ brain activity changes in response to fatigue [10,11]. In this regard, Lim et al. [12] found that individuals exhibit considerable variability in specific EEG brain waves, presenting differentiations in power during prolonged cognitive tasks, with some showing rapid onset of fatigue indicators and others maintaining stable EEG patterns over time. Furthermore, Borghini et al. [13] observed significant differences in how fatigue affected brain connectivity, with some drivers showing marked reductions in the fronto-parietal network, while others exhibited minimal changes.

As such, the development of objective mental fatigue estimation systems is crucial to enhance road safety. This, combined with current developments in technology such as mobile sensors, can mitigate the risk of accidents due to impaired driving performance, promptly alerting drivers to their cognitive exhaustion level [14,15]. Specifically, regarding the development of wearable EEG technologies, considerable research efforts have been directed towards overcoming limitations and facilitating the long-term, non-invasive recording of brain signals during individuals’ mobility outside laboratory settings [16]. Some solutions have focused on advancements in the development and refinement of materials and techniques for creating stretchable circuits, which include methods like mask deposition, laser patterning, and printing methods [2]. Concurrently, other studies have concentrated on reducing the number of electrodes to aid developers of future EEG applications in choosing the optimal electrode positions [17].

To further enhance the efficiency of mental fatigue detection, recent studies have performed band specific identification of the alterations in brain function. Different brain wave frequencies, such as theta, alpha, beta, delta, and gamma, have been associated with various cognitive states and processes, including fatigue [18,19]. As such, focusing on specific brain waves in fatigue detection can provide more targeted and nuanced assessments of fatigue levels [20]. Although, several studies propose that all frequency bands could potentially be utilized in identifying driving fatigue [21,22], the majority suggest that measures within the theta, alpha, and beta bands are more closely associated with it, with fewer studies referencing delta or gamma activity as potential contributors [23,24]. In addition, analysis of brain operations as having a structured organization can highlight detailed patterns, comprising distinct regions that collaboratively interact to establish expansive distributed networks [25]. These networks pertain to collections of cerebral regions that contribute to the execution of specific interconnected tasks or a designated array of functions, while quantifying statistical similarities in brain activity, thereby revealing the intricate neural processes [26,27]. The differences within the brain networks have also been examined to distinguish between different cognitive states. In fact, Tompson et al. [28] conducted a review on the temporal dynamics of dynamic brain networks, noting that variations in both the strength and adaptability of evolving brain networks play a role in shaping individual disparities in executive function, attention, working memory, and learning abilities. In another study, Li et al. [11], employed a long-term driving EEG task, reporting significantly increased patterns in functional connectivity in theta, alpha, beta, and gamma bands, suggesting a compensation effect to attenuate the effects of driving fatigue.

Human EEG data research traditionally concentrates on the analysis of data aggregated across groups, a practice that constrains the level of detail, specificity, and clinical applicability of task-based functional connectivity maps [29]. However, despite the considerable volume of relevant research, the inter-subject variability of the brain networks during different tasks and cognitive states renders universal approaches inconclusive. In fact, Andrew James et al. [30] employed a comparative analysis of two methodologies for constructing a human brain atlas applicable to both resting-state and task-state conditions. Their findings demonstrated that an individualized approach yielded more statistically significant results. In addition, Fin et al. [31] illustrated that distinct functional connectivity profiles serve as unique “fingerprints”, facilitating precise identification of subjects within a sizable cohort. They proposed that an individual’s connectivity profile is inherent and can effectively distinguish them irrespective of the brain’s state during imaging. Furthermore, Tavor et al. [32] suggested that individual variances in brain responses are predominantly intrinsic to the brain and can be anticipated based on task-independent measurements obtained during rest. By employing a diverse range of task conditions spanning multiple domains, they predicted individual differences in brain activity and revealed a link between brain connectivity and function that can be captured at an individual subject level. Moreover, Sun et al. [33] employed a combination of individualized prediction models with quantitative graph theoretical analysis across schizophrenia symptom categories. Their analyses of individual-specific functional connectivity provided significant distinctions among cortical circuits linked to positive and negative symptoms, shedding light on how circuits underlying symptom manifestation may differ depending on the underlying cause of the illness.

Although fatigue states exhibit notable differences compared to rested states, suggesting significant disparities [34], sustained attention during long-term driving does not always display monotonic development trends [35]. This is often attributed to factors irrelevant to mental fatigue such as shifts of attentional control induced by additional activities or unrelated visual and auditory stimuli [36].

Taking the above into consideration, we hypothesized that assessing monotonic alterations in brain activation would eliminate extraneous cognitive procedures and therefore enable the assessment of the neuronal connections specific to mental fatigue, investigating individual subject deviations. As such, we employed an EEG sustained driving simulation experiment, while conducting a linear regression analysis of the entire duration of the experiment, incorporating individual distinct task considerations. In this regard, we utilize an individual participant network analysis based on the Phase Lag Index (PLI), to enhance the validity of our findings. This amalgamation yields a comprehensive network capturing shared connectivity patterns across the entire participant group, providing a collective snapshot of neural dynamics. Moreover, by focusing on the common connections of the EEG-derived brain networks, we provided indications for feature space minimization, therefore allowing for mobile sensor-based effective fatigue detection in real-time.

## 2. Materials and Methods

### 2.1. Participants

In this study we recruited 21 individuals (average age 25.2 ± 6.1 years, all right-handed) from the National University of Singapore (NUS), including students and staff members. All participants held valid driver’s licenses and had either normal or corrected-to-normal vision. Before the experiment, participants completed a self-administered questionnaire to ensure they met inclusion criteria, which included the absence of fatigue-related disorders and chronic physical or mental illnesses, no long-term medication usage, avoidance of caffeine or alcohol consumption, and obtaining more than 7 h of sleep in the two days preceding the experiment. Individuals who did not meet these criteria were excluded. Prior to the start of the experiment, all participants provided informed written consent, and they were compensated SGD 10 per hour for their participation. The research protocol was approved by the Institutional Review Board of NUS.

### 2.2. Experimental Design

The experimental procedure was designed to elicit mental fatigue based on the premise that sustained attention while driving (even with low demands for complex cognitive processing) can lead to monotony, a significant contributor to mental fatigue. Relevant studies have indicated that performance and alertness can decline significantly even within a relatively short period, such as 20–30 min, due to the repetitive and unstimulating nature of the task [37]. In this regard, participants operated the driving simulation for 1 h, using a driving wheel, pedals, and a gear box (Logitech G27 Racing Wheel). The unvaried route chosen for the task encompassed both a motorway and a rural road, primarily characterized by straight sections with minimal traffic, aimed to induce a state of drowsiness among the participants. Participants received instructions to maintain stable driving conditions, with a maximum speed limit of 100 km/h. The simulated driving task involved the utilization of City Car Driving software (Version 1.5, http://citycardriving.com/) with adherence to Singapore’s driving rules. To minimize participant movements and consequently reduce the potential for significant electromyography (EMG) artifacts, an automated clutch system was implemented. A visual representation of the experimental design can be seen in Figure 1.

### 2.3. Behavioral Assessment

To evaluate if the experimental procedure was effective in inducing mental fatigue, the Short Stress State Questionnaire (SSSQ) was filled out by each participant [38]. Briefly, the SSSQ is designed to evaluate the subjective state of individuals in stressful environments. It comprises 24 questions that measure three factors: Engagement (assessing the level of arousal, motivation, and concentration necessary for task completion), Distress (encompassing unpleasant mood and tension, alongside a lack of confidence stemming from perceived lack of control over work), and Worry (involving cognitive aspects such as attention, self-esteem, and cognitive interference), with each factor being assessed through 8 specific questions. The SSSQ was applied twice, before and after the driving simulation session.

### 2.4. EEG Acquisition and Pre-Processing

During the driving simulation experiment, EEG data were collected employing 64 Ag/AgCl scalp electrodes (Waveguard from ANT B.V., Hengelo, The Netherlands) following the standard 10–20 system [39]. Data collection was implemented at a sampling rate of 512 Hz, while maintaining electrode impedance levels below 10 kOhms throughout the entire recording process. Common sources of interference were diligently addressed by applying a band-pass filter (0.5–70 Hz) and a 50 Hz notch filter. Concurrently, bipolar horizontal and vertical electrooculogram (EOG) signals were recorded to detect eye blinks that usually result in EEG artifacts [40]. A previously validated EEG pre-processing pipeline was utilized that included down-sampling, filtering, re-referencing, and artifact removal [41]. Specifically, the raw EEG data were down-sampled to 256 Hz, re-referenced to the average of electrodes located on the left and right mastoid, and band-pass (FIR) filtered within the frequency range of 1 to 45 Hz. Then artifact removal was applied with the use of Independent Components Analysis (ICA) to identify and eliminate components exhibiting high correlation with EOG signals. Baseline correction involved estimation and removal of baseline fluctuations of the entire duration of each epoch. To further ensure high data quality, continuous data quality control measures were enacted by excluding data segments manifesting power levels exceeding 6 decibels (dB) within high-frequency bands (20–40 Hz). All data pre-processing procedures were implemented in MATLAB Version R2022b (Mathworks Inc., Natick, MA, USA) through the EEGLAB tool [42].

### 2.5. Functional Connectivity Based on PLI Networks

For the purpose of connectivity estimation, the time-series data were systematically partitioned into consecutive 5 min intervals, each with a 50% temporal overlap. To eliminate subjective phenomena relevant to experimental engagement, the first and the last 5 min intervals were discarded, resulting in 20 overall EEG epochs (Figure 1c). Subsequently, a Phase Lag Index (PLI) network was calculated for each participant across each epoch in different frequency intervals (i.e., delta band (0.5–4 Hz) theta band (4–7 Hz), alpha band (8–12 Hz), beta band (13–30 Hz), and gamma band (31–45 Hz)) as described below.

The selection of PLI for EEG connectivity analysis was motivated by its ability to assess the phase synchronization in EEG signals, while minimizing false positives due to volume conduction, its robustness against noise and artifacts, and its focus on capturing true functional connectivity with non-zero phase lag [29,43].

This is accomplished by discounting phase differences (angles) of zero and π radians. PLI quantifies the asymmetry in the distribution of instantaneous phase differences, which are determined through the application of the Hilbert transformation. Specifically, for any arbitrary EEG signal x(t), the analytic signal w(t) is constructed with a complex function of time:(1)$$w\left(t\right)=x\left(t\right)+{\mathrm{ix}}_{H}\left(t\right)=x\left(t\right)+{i\pi }^{-1}\mathrm{pv}\int_{-\infty }^{+\infty }\frac{x\left(s\right)}{t-s}\mathrm{ds}$$
where $x_{H}\left(t\right)$ is the Hilbert transform of c(t), and pv denotes the Cauchy principal value. The Hilbert transform is virtually the convolution of x(t) with 1/πt.

The instantaneous amplitude A(t) and instantaneous phase ϕ(t) of x(t) can be derived in terms of the analytic signal’s polar form:(2)$$w=A\left(t\right){\;e}^{i\varphi \left(t\right)}$$

In turn, the phase is uniquely determined as follows:(3)$$\varphi \left(t\right)=\arctan\;(\frac{x_{H}\left(t\right)}{x\left(t\right)})$$

From the phases of two EEG signals x_a_(t) and x_b_(t), the phase difference or relative phase is formed as follows:(4)$$\varphi_{\mathrm{ab}}\left(t\right)={\;\varphi }_{\alpha }\left(t\right)-\varphi_{b}\left(t\right)$$

Then, the PLI is defined as an asymmetry measure for the phase difference distribution using the following equation:(5)$${\mathrm{PLI}}_{\mathrm{ab}}=\left|\frac{1}{N}\overset{N-1}{\underset{n=0}{\sum }}\mathrm{sign}\left(\varphi_{\mathrm{ab}}\left(n\right)\right)\right|$$

A symmetric distribution, centered around zero, may indicate spurious connectivity, while a flat distribution signifies the absence of connectivity. Departures from symmetry indicate interdependencies among sources. PLI values are bounded between 0 and 1. A value of 0 signifies either no coupling or coupling with phase differences predominantly centered around zero, whereas a value of 1 indicates precise phase locking at a non-zero phase difference. PLI values in proximity to 1 suggest robust non-zero phase locking. The extracted networks are represented as weighted undirected triangular adjacency matrices (with the dimensions in this paper being 62 × 62). As mentioned above, in order to facilitate a comprehensive exploration of functional connectivity dynamics across distinct frequency bands, the PLI functional brain networks were computed within the spectral domains of the delta, theta, alpha, beta, and gamma bands.

### 2.6. Individual and Global Network Analysis

Within the scope of our study, we implemented an individual participant network analysis (IN) and a Global Network analysis (GN). These terminologies aim to bring clarity to the distinction between analyses conducted at the group level, where EEG-derived brain networks are collectively examined, and analyses undertaken at the individual level, focusing on the personalized EEG-derived brain networks for each participant. Specifically, the IN entailed analyzing EEG data for each participant separately, constructing unique brain networks customized to the intricate features of each individual’s neural patterns. As such, IN included the 5 min time interval network of each participant for each frequency band, to illuminate subject-specific variations.

Conversely, the GN analysis involved the aggregation of EEG-derived brain networks from all participants. This amalgamation yields a comprehensive network capturing shared connectivity patterns across the entire participant group, providing a collective snapshot of neural dynamics. In this regard, the GN analysis included the construction of an average grid, calculated as an average of individual PLI networks (i.e., determining the mean value of the interconnected edges of all participants) for each frequency band.

### 2.7. Network Assessment

As mentioned above, the PLI networks were computed within each participant’s dataset to quantify phase relationships among the 62 scalp electrodes’ signals. To ascertain the significance of the network edges, we conducted a linear regression analysis, systematically assessing the relationships between all possible pairs of sensors. A flowchart of the analysis framework is presented in Figure 2. Briefly, linear regression serves as a statistical modelling tool for examining and quantifying the relationships in complex systems [44]. This entails fitting a linear equation to observed data points, aiming to elucidate the underlying pattern governing the association between variables. The fitting line represents the optimal mathematical approximation (defining the best-fitting line) that minimizes the overall difference between observed data points and their corresponding values, determining the coefficients of the equation (such as the slope and intercept). The optimization process involves minimizing the sum of squared differences between observed and predicted values, employing the least squares method. Moreover, to provide insights into the strength, direction, and significance of network relationships, a hypothesis evaluation was implemented, testing the null hypothesis that the independent variable has no correlation with the dependent variable. Linear regression was performed by fitting a linear model to the data by minimizing the sum of the squared residuals. The general form of the model was as follows:(6)$$y=\beta_{0}+\beta_{1}x_{1}+\beta_{2}x_{2}+\cdots +\beta_{k}x_{k}+\epsilon$$
where *y* represents the dependent variable and x_1_, x_2_, …, x_k_ are the independent variables, β_0_, β_1_, …, β_k_ are the regression coefficients, and *ϵ* is the error term.

This function estimates the coefficient β of the linear model using the Ordinary Least Squares (OLS) method:(7)$$\beta ={\left({Χ}^{Τ}Χ\right)}^{-1}{Χ}^{Τ}y$$
where X is the matrix of independent variables, and y is the vector of the dependent variable.

As mentioned in Section 1, the rationale for this procedure stems from evidence indicating that fatigue states exhibit notable dissimilarities compared to rested states, thus implying statistical differences [34]. In addition, during sustained fatigue-inducing tasks, cognitive drain (excluding irrelevant stimuli) is expected to increment continuously [45]. Consequently, as mental resources progressively deteriorate, the corresponding neural connections (in our study the PLI network edges) should demonstrate a discernible deviation from previous states, following an increasing or decreasing trajectory. It is important to note that the patterns analyzed here do not imply that mental fatigue dynamics present a linear behavior, since similar studies denote nonlinear trends [46]. However, linear regression can elucidate the way (driving) mental fatigue affects brain connectivity over time, identify critical brain networks affected, and help us understand individual differences in susceptibility to fatigue.

Accordingly, the 62 × (62 − 1)/2 = 1891 unique connections were calculated across 20 epochs of PLI networks. Subsequently, linear regression analysis was conducted on each connection (treating PLI values as dependent variables and time intervals as independent variables) to assess an increasing or decreasing trend of the PLI connection weights. In this study, a critical threshold for connection strength was established utilizing the R-squared (R^2^) statistical measure [47]. R^2^ is a measure that represents the proportion of the variance for a dependent variable. As such, it provides an indication of how well data points fit a statistical model. In this study, an R^2^ value greater than 0.25 corresponds to a *p*-value less than 0.05 (*p* < 0.05), denoting less than a 5% probability that the observed correlation occurred by chance. By setting these thresholds (R^2^ > 0.25 and *p* < 0.05), we ensured that only connections with a certain level of explanatory power and statistical significance were considered meaningful for subsequent analysis. Connections exhibiting correlations surpassing this threshold were considered highly significant and were retained for further examination. In both IN and GN, we estimated time-dependent significant changes, by utilizing the R^2^ and *p* value criteria between the different 5 min epochs, to assess the connections that exhibited significant differences throughout the entirety of the driving simulation.

## 3. Results

### 3.1. Behavioral Results

To investigate the subjective effect of the experimental design with regards to mental fatigue, we statistically assessed the SSSQ scores. As such, we applied a One-Way Analysis of Variance (ANOVA) comparing pre-task and post-task Engagement scores. ANOVA revealed a significant difference (*p* < 0.01) indicating the experimental design was effective in inducing mental fatigue.

### 3.2. Individual Participant Network Analysis

Following the threshold assessment of the linear regression procedure, the significant connections (across the 5 min PLI networks for each frequency band) were investigated in terms of shared attributes between the different subjects. Interestingly, no connections exhibited significant changes common to all subjects across any of the frequency bands studied. Further analysis indicated a consistent pattern, where the significant changes across all time frames encompasses roughly 50% of the subjects (10 out of 21). This was consistent in all frequency bands, with no significant edges being observed for over half of the subjects, where two shared connections were detected in 50–60% of individuals (Figure 3). In this regard, we arbitrarily decided to further analyze the PLI edges that were present for at least 40% of the subjects. Figure 4 displays the common connections shared over the 40% threshold for each frequency band).

Concerning the distinct pairs of electrodes within the alpha band, a collective count of 33 PLI connections was observed across 40 unique nodes (Figure 4c). Notably, a considerable majority of connectivity changes were intra-hemispheric. Further investigation in all electrode positions presented a left hemisphere dominance, with a relatively lower occurrence in the right areas. Within the theta band, 21 connections were identified across 22 unique nodes (Figure 4b), with a significant proportion (>50%) of the observed connections being localized to electrodes positioned along the midline, spanning both hemispheres. In the beta band, a cumulative total of 11 connections was noted across 20 unique nodes (Figure 4d), where the majority of connections were constrained within the same hemisphere. In both the delta (10 connections were observed across 17 unique nodes) and gamma bands (9 connections were observed across 16 unique nodes) (Figure 4a,d), a noteworthy spatial tendency was observed. Specifically, a substantial proportion of electrode connections in these frequency bands predominantly involve the right hemisphere.

To further assess the connection alterations, we examined the slopes obtained via the linear regression approach, offering a focused insight into the overall increases or decreases in particular connections. Specifically, under the >40% threshold, a pattern could be discerned within the alpha band, with the majority (24 out of 33) of PLI edges showing an upward slope. Conversely, within the theta band, a downward inclination of the regression line was displayed in 18 out of 21 connections. Interestingly, in all beta, gamma, and delta frequency bands, approximately half of the connections demonstrate a negative and the remaining half a positive slope (beta: six negative/five positive; gamma: five negative/four positive; delta: four negative/six positive).

### 3.3. Association of Individual to the Global Network

To estimate the interrelationships between subject variability and the average brain network (pertaining mental fatigue), the GN was also examined with the same R^2^ and *p* value criteria (i.e., *p* < 0.05 and R^2^ > 0.25). The number of connections exhibiting significant alterations for delta, theta, alpha, beta, and gamma bands is presented in Figure 5 below.

Remarkably, the delta band demonstrated the largest count of connections (440) exhibiting significant alterations across our experimental conditions, while the gamma band exhibited the least number of connections (Table 1).

Subsequently, the association between the GN and IN was investigated. As such, we compared the IN to the GN, to establish the association between average and subject-specific networks. Specifically, the connections exhibiting significant alterations were compared to the average network, calculating the number of PLI edges that were included in both the IN and GN (Table 2). From this perspective, subject variability was further emphasized with different individuals illustrating various levels of shared connections. In detail, the highest number of shared connections were included in the alpha band (21.5%), ranging from individuals with 7.8% to 47.37% common connections between the IN and GN. On the contrary, the gamma band displayed the lowest number of mutually shared connections (16.3%), nevertheless demonstrating significant range (7.8% to 36.95%).

## 4. Discussion

In this study we aimed to illustrate the intricacies of subject variability in brain connectivity with respect to driving fatigue. As such, we implemented a comprehensive analysis focusing on subject-specific brain networks derived via a linear regression method. Our analysis elucidated the distribution of subject-specific slopes, revealing insights into the variability and trends in connectivity dynamics specific to each frequency band. These indicators provide valuable insights into the nuanced relationships between frequency and connectivity patterns in the context of driving fatigue. In addition, our approach addresses the variations in individual brain networks, thereby enhancing the accuracy of fatigue assessment by considering these variations rather than disregarding them. Moreover, the identification of a small subset of distinctive network edges holds significant advantages for mobile sensor-based applications, particularly in the context of efficiency, resource optimization, and real-time processing.

As previously stated, we employed linear regression analysis in the PLI network edges to estimate significant differences of a monotonic fashion. As such, we hypothesized that sustained attention in driving would consistently allocate and deplete finite cognitive resources, inevitably inducing mental fatigue in a continuous manner [48]. However, recent studies report non-monotonic fatigue-related development trends at the individual level [10,49]. This is attributed to fatigue adaptation or self-regulation (involving and implementing strategies and behaviors to prevent or mitigate the negative effects of fatigue), optimizing performance amidst conditions that induce fatigue [50,51]. Although this is the case in real-word scenarios (shifting attentional control induced by additional activities, such as advanced road familiarization, listening to music or media, adjusting windows, readjusting sitting position, etc.) [36], our study aimed to assess mental fatigue, only focusing on experimental factors and thus excluding non-fatigue-related operations. From this standpoint, by exclusively estimating significant connections that exhibit an increasing or decreasing trend over the entire duration, we can avoid incorporating fatigue characteristics that could be internally regulated [11].

A key finding from our study underscores the efficacy of subject-specific networks for analyzing fatigue-related brain dynamics, undergoing alterations during the fatigue-inducing driving task. This was indicated in all frequency bands examined, corroborated with functional connectivity driving-fatigue studies [52,53]. Regarding the shared connections between all subjects, the alpha band displayed the highest sensitivity to the driving simulation. This is supported by other studies, reporting significant alterations in the alpha frequency band [54]. The analysis of the common (in >40% of subjects) PLI connections presented a left lateralization that could potentially be indicative of specific cognitive processes associated with fatigue, possibly related to attention and alertness [55]. Conversely, the right hemisphere exhibits a comparatively lower incidence of alterations, suggesting a hemisphere-specific response to driving-induced fatigue [10]. Interestingly, the majority of the significant alpha band edges where interhemispheric, a pattern which is in agreement with similar studies [56]. In terms of theta band shared connections, the majority were located in the frontal and prefrontal area of the brain, aligning with previous research findings [18,57,58]. Further analysis revealed a notable prevalence of connections along the midline regions of the brain. This prominence underscores their potential role in mediating cognitive processes (such as attention and executive function), suggesting a specific sensitivity or susceptibility of midline structures to changes in fatigue-induced connectivity [59]. The beta band exhibits a distinct trend with alterations primarily confined within the same hemisphere. This observation emphasizes the potential regional specificity inherent in beta frequency alterations, suggesting that changes in beta oscillations may be more concentrated within specific brain regions or networks [43,60]. Furthermore, in the context of driving-induced fatigue, it indicates that the functional reorganization of the brain, particularly within specific hemispheres, could play a critical role in modulating neural activity associated with fatigue [61]. The distinctive spatial preference detected within the delta and gamma frequency bands (with a pronounced impact on the right hemisphere), signifies a specific pattern of connectivity adjustments. This finding underscores the intricate interplay between neural networks of low- and high-frequency brain dynamics, shedding light on how hemispheric differences contribute to overall connectivity and its implications for cognitive processes and behaviors, particularly in contexts such as driving fatigue, where precise neural coordination is paramount [62,63].

In the GN analysis, it becomes apparent that the delta band exhibits the highest proportion of significantly altered connections. This could indicate drowsiness, consistently associated the delta band, which is a well-documented phenomenon frequently observed during extended periods of driving [64,65,66]. Notably, however, this correlation between delta band activity and drowsiness is not obvious in the analysis conducted in the IN. This intriguing finding underscores the considerable subject variability in the experience of drowsiness during driving experiments, thus shedding light on the diverse effects of individuality within the scope of this study.

Further investigation of the individualized connectivity associated with the GN also revealed subject-specific complexities of fatigue assessment in the context of driving fatigue. As such, the average number of network connections reflecting each individual’s brain network (1/5 of the total connections) shows variations in the individual network constructions. This is in line with other relevant research works that state that average networks in functional connectivity analyses may fail to reveal actual independent human brain network procedures [67,68]. In fact, recent studies have highlighted the substantial magnitude of cross-individual variation in functional connectivity strength, surpassing variations observed within subjects across different states [69]. This emphasizes the concept that variability in functional connectivity functions as a metric at the trait level, adept at correlating with various trait-level measures across individuals. Moreover, a substantial body of research substantiates the notion that these variances are indeed associated with individual disparities in cognitive functions [70,71].

The interplay between the IN and GN is crucial in understanding the neural basis of driving fatigue. Mental fatigue impacts both localized brain activities and large-scale neural interactions, disrupting the functional integration necessary for effective driving [72]. Within this study, the subject variability observed when analyzing the IN patterns and their associations with the GN (with some participants displaying significant decreases in connectivity, while others maintain more stable patterns) suggests that fatigue impacts brain connectivity differently across individuals. Some individuals might employ adaptive strategies to cope with fatigue (such as using mental techniques to maintain alertness), thus contributing to individual differences in fatigue susceptibility and neural responses [73]. Future work can offer valuable insights into the neural mechanisms underlying subject-specific driving fatigue and develop interventions to mitigate its effects.

Taken the above into consideration, it can be inferred that individual-specific analysis could enhance driving-fatigue detection utility. By prioritizing the most relevant network edges, mobile sensor-based applications can swiftly extract meaningful patterns or anomalies from sensor data, enabling rapid responses to changing conditions or events. Moreover, the expedited data transmission and analysis resulting from focusing on a small subset of network edges can contribute to faster insights and actionable information. These notions, paired with network metrics (due to the small wordless characteristics in fatigue states [27]) could significantly enhance the development of more effective and resource-efficient applications that can address a wide range of real-world challenges.

Certain limitations have to be taken into consideration when interpreting the results of this study. Firstly, our study incorporated exclusively male subjects. This was implemented due to the consistently reported existence of sexual dimorphism in brain structures that could contribute to variations in behavior and cognitive function [74]. By only enrolling male participants we aimed to investigate the intrinsic impact of mental fatigue, independent of potential gender-related influences. Another important limitation is the relatively small sample size (albeit carefully selected), that may pose constraints on the generalizability of the findings. Further research with larger and more diverse samples, incorporating real-world driving conditions, will be essential to validate and extend our findings.

## 5. Conclusions

This study focuses on the significance of subject-specific brain networks in understanding connectivity dynamics related to driving fatigue. These findings highlight the variability in individual brain networks across different frequency bands, shedding light on the relationships between frequency and connectivity patterns in the context of fatigue assessment. By focusing on distinct network edges and employing linear regression analysis, this study highlights the potential of subject-specific networks in analyzing fatigue-related brain dynamics. Moreover, the identification of shared connections between participants provides insights for feature space minimization, laying the groundwork for effective fatigue detection using mobile sensor-based applications.

## Figures and Tables

**Figure 1 sensors-24-03894-f001:**
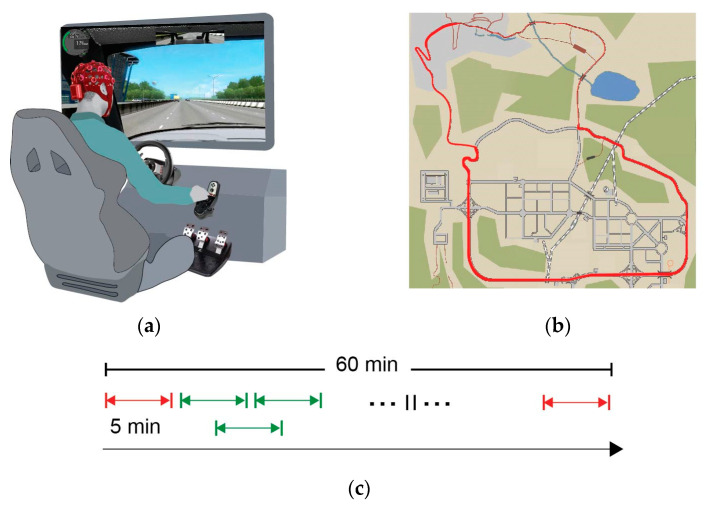
A schematic of the experimental design: (**a**) the driving simulation setup; (**b**) the predefined route over the simulation map is presented with red; and (**c**) the duration of the driving simulation was 1 h, continuously recording EEG. In the subsequent analysis, EEG was divided in 5 min segments with 50% overlap (green), while segments that corresponded to the first and last 5 min (red) were excluded.

**Figure 2 sensors-24-03894-f002:**
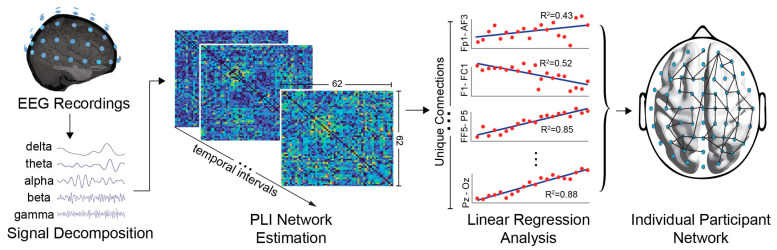
The outline of the adopted framework. The PLI networks are calculated based on the frequency bands deriving from the EEG signal. Then, the PLI edges are examined based on the R^2^ measures, delineating increasing or decreasing patterns and their significance level to further analyze the resulting networks.

**Figure 3 sensors-24-03894-f003:**
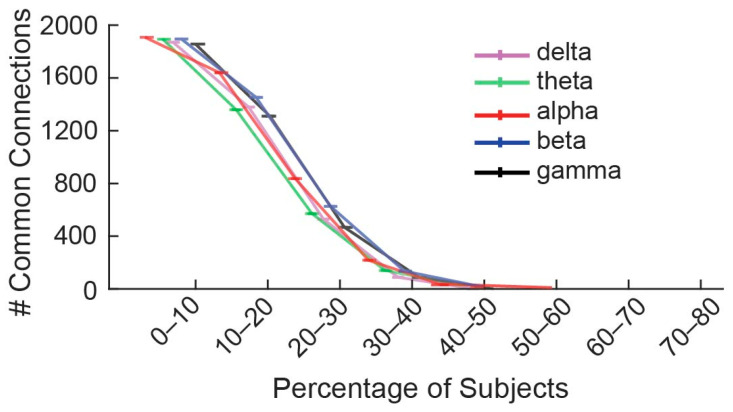
Visual representation of the percentage range of subjects that share common connections.

**Figure 4 sensors-24-03894-f004:**
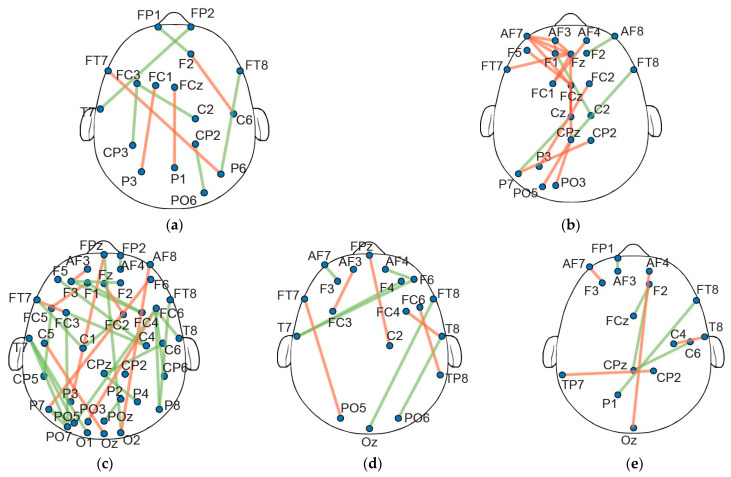
The significant IN connections for (**a**) delta band; (**b**) theta band; (**c**) alpha band; (**d**) beta band; and (**e**) gamma bands. Connections colored green display positive slope, while connections with orange display negative slope.

**Figure 5 sensors-24-03894-f005:**
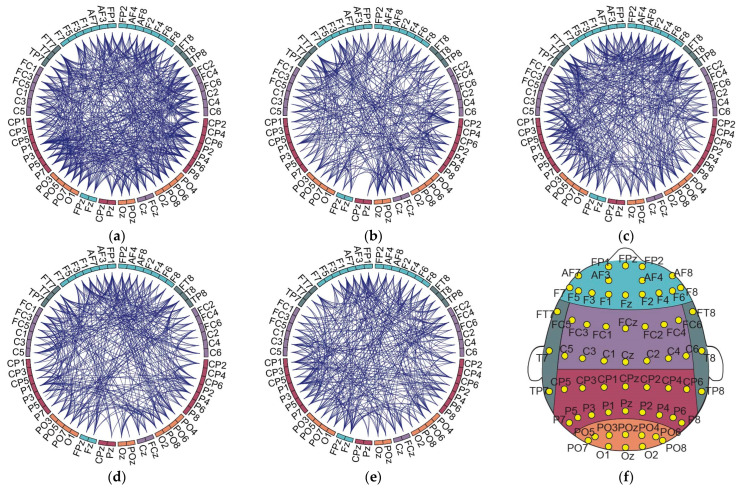
Significant connections in the GN for (**a**) delta; (**b**) theta; (**c**) alpha; (**d**) beta; and (**e**) gamma frequency bands and (**f**) the electrode positions (color-coded according to the circular diagrams) for frontal, temporal, central, parietal, and occipital scalp areas.

**Table 1 sensors-24-03894-t001:** Number of significant connections (*p* < 0.05 and R^2^ > 0.25).

Frequency Band	Number of Connections
delta	440
theta	287
alpha	361
beta	313
gamma	275

**Table 2 sensors-24-03894-t002:** The shared connections between the IN and GN.

Frequency Band	Mean Percentage ^1^ (%)	Min Percentage (%)	Max Percentage (%)
delta	17.5 ± 6	10.00	29.09
theta	18.3 ± 7	9.76	32.06
alpha	21.5 ± 11	7.8	47.37
beta	18.4 ± 6	10.54	35.14
gamma	16.3 ± 7	7.80	36.95

^1^ The standard deviation is presented next to the mean percentage value after the ± symbol.

## Data Availability

The data presented in this study are available on request. The data are not publicly available due to privacy reasons.

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
