# Peer review of "Individual Variability in Brain Connectivity Patterns and Driving-Fatigue Dynamics"

_sensors, 2024, doi:10.3390/s24123894_

Round 1

Reviewer 1 Report

Comments and Suggestions for Authors

In this study, the authors attempt to identify patterns of cortical connectivity networks in subjects subjected to a state of fatigue. These states were evoked by the activity of driving a car for one hour. My first question is: how did the authors ensure the presence of the state of fatigue in the subjects? Experimentally, it would be expected that the subjects enter the state of fatigue at different times, so I agree with the authors that the analysis should be addressed for each individual separately.

The analysis methodology employed by the authors is primarily based on the assumption stated between lines 238 and 240, which reads: “Consequently, as mental resources progressively deteriorate, the corresponding neural connections (in our study the PLI network edges) should demonstrate a discernible deviation from previous states, following a linear trajectory.” This assumption, in my opinion, should be demonstrated. The authors have not presented any empirical evidence to support it.

Does this imply that the dynamics by which the subject moves towards the state of fatigue are linear? This is an ideal that needs to be demonstrated. From my point of view, demonstrating that this temporal dynamic in connectivity patterns exhibits linear behaviors representing a relative state of fatigue would be a significant discovery (if it has not already been demonstrated).

This brings me to my first question: how can the authors ensure that the subjects have reached or have not reached the state of mental fatigue?

The introduction section states that certain complex cognitive states, such as imagination, fatigue, and others, present individual patterns in each subject, which makes the attempt to generalize systematic global patterns (between subjects) very complicated. This is correct; however, the authors should delve more specifically into the cortical variabilities that other studies have found in states of fatigue. In this context, a more specific definition of "mental fatigue" needs to be provided in this work.

The work is well-structured; however, its working hypothesis needs to be better justified. This is essential for the validity of the results they have obtained.  

Reviewer 2 Report

Comments and Suggestions for Authors

The manuscript presents an electroencephalography (EEG)-based fatigue driving detection system with phase lag index (PLI) for analyzing individual network (IN) and global network (GN). There are some areas that could be further improved.

1. How to identify if the driver is fatigued using the proposed system? Additionally, how to label fatigue driving? by questionnaire or other relevant labeling methods?

2. The paper seems too descriptive and somehow diluted. I recommend the paper to get mathematical style in Sections 2.4-2.6.

3. The authors should add some text for explanations about the relevance of PLI for fatigue driving. Additionally, why select PLI?

4. The authors should add R-square values in Figure 2 and add some text for explanations about these values between PLI.

5. The authors should add some explanations and citations for text “critical threshold for connection strength was established at R-squared, r> 0.25, corresponding to a p-value p < 0.05” on page 6, L 249 to L 250.

6. Could the authors calculate average values, minimum values, and maximum values of the shared connections between IN and GN in Table 2, respectively? Additionally, the authors should add some text for explanations about the relationships between both and fatigue driving.

Round 2

Reviewer 1 Report

Comments and Suggestions for Authors

I am satisfied with the response the authors have provided to my main concern. It is true that linear behavior in any type of variable related to cortical dynamics evoked by complex cognitive states could be a first approximation. In this regard, the fact that the subjects have reached certain levels of fatigue can be quite relative; however, the path towards this state is what is being analyzed. The authors have adequately justified this approach in the text. Based on this, and the previous review, I can recommend this work for publication in this journal.

Reviewer 2 Report

Comments and Suggestions for Authors

All the comments are addressed in the manuscript by the authors.